# Differences in IgG Antibody Responses following BNT162b2 and mRNA-1273 SARS-CoV-2 Vaccines

José G. Montoya,[a] Amy E. Adams,[b] Valérie Bonetti,[a] Sien Deng,[c] Nan A. Link,[b] Suzanne Pertsch,[b] Kjerstie Olson,[a] Martina Li,[c] Ellis C. Dillon,[c] Dominick L. Frosch[c]

aDr. Jack S. Remington Laboratory for Specialty Diagnostics, Palo Alto Medical Foundation Research Institute, Palo Alto, California, USA
bPalo Alto Foundation Medical Group, Palo Alto, California, USA
cCenter for Health Systems Research, Sutter Health, Palo Alto Medical Foundation Research Institute, Palo Alto, California, USA

**ABSTRACT** Studies examining antibody responses by vaccine brand are lacking and may be informative for optimizing vaccine selection, dosage, and regimens. The purpose of this study is to assess IgG antibody responses following immunization with BNT162b2 (30 $\mu$g mRNA) and mRNA-1273 (100 $\mu$g mRNA) vaccines. A cohort of clinicians at a nonprofit organization is being assessed clinically and serologically following immunization with BNT162b2 or mRNA-1273. IgG responses were measured at the Remington Laboratory by an IgG assay against the SARS-CoV-2 spike protein-receptor binding domain. Mixed-effect linear (MEL) regression modeling was used to examine whether the SARS-CoV-2 IgG level differed by vaccine brand, dosage, or number of days since vaccination. Among 532 SARS-CoV-2 seronegative participants, 530 (99.6%) seroconverted with either vaccine. After adjustments for age and gender, MEL regression modeling revealed that the average IgG antibody level increased after the second dose compared to the first dose ($P < 0.001$). Overall, titers peaked at week 6 for both vaccines. Titers were significantly higher for the mRNA-1273 vaccine on days 14 to 20 ($P < 0.05$), 42 to 48 ($P < 0.01$), 70 to 76 ($P < 0.05$), and 77 to 83 ($P < 0.05$) and higher for the BNT162b2 vaccine on days 28 to 34 ($P < 0.001$). In two participants taking immunosuppressive drugs, the SARS-CoV-2 IgG antibody response remained negative. mRNA-1273 elicited higher IgG antibody responses than BNT162b2, possibly due to the higher S-protein delivery. Prospective clinical and serological follow-up of defined cohorts such as this may prove useful in determining antibody protection and whether differences in antibody kinetics between the vaccines have manufacturing relevance and clinical significance.

**IMPORTANCE** SARS-CoV-2 vaccines using the mRNA platform have become one of the most powerful tools to overcome the COVID-19 pandemic. mRNA vaccines enable human cells to produce and present the virus spike protein to their immune system, leading to protection from severe illness. Two mRNA vaccines have been widely implemented, mRNA-1273 (Moderna) and BNT162b2 (Pfizer/BioNTech). We found that, following the second dose, spike protein antibodies were higher with mRNA-1273 than with BNT162b2. This is biologically plausible, since mRNA-1273 delivers a larger amount of mRNA (100 $\mu$g mRNA) than BNT162b2 (30 $\mu$g mRNA), which is translated into spike protein. This difference may need to be urgently translated into changes in the manufacturing process and dose regimens of these vaccines.

**KEYWORDS** COVID-19, SARS-CoV-2, BNT162b2, mRNA-1273, vaccine, IgG, antibodies, vaccines

Within 1 year of the emergence of SARS-CoV-2, two novel and effective mRNA vaccines became available, BNT162b2 (Pfizer/BioNTech) and mRNA-1273 (Moderna) (1, 2). Thirty micrograms of BNT162b2 mRNA is translated into the SARS-CoV-2 full-

Address correspondence to José G. Montoya, montoyj@sutterhealth.org.

**TABLE 1** Demographics of 611 participants who completed baseline assessment and serum

| Characteristic | N | % |
|---|---|---|
| Gender | | |
| Female | 410 | 67.1 |
| Male | 201 | 32.9 |
| Age (mean, 47.4) | | |
| 28–39 | 140 | 22.9 |
| 40–49 | 229 | 37.5 |
| 50–59 | 161 | 26.4 |
| 60–76 | 81 | 13.3 |

length spike protein (prefusion conformation) and boosted 3 weeks after (3). One hundred micrograms of mRNA-1273 mRNA is translated into the prefusion-stabilized spike glycoprotein and boosted 4 weeks later (4).

Healthcare workers were the first group to receive BNT162b2 and mRNA-1273 (1). The present study was launched on 10 December 2020, the week that SARS-CoV-2 vaccines became available, providing the opportunity to assess antibody responses in participants receiving two different vaccine brands, before and after immunization. Most studies so far have focused on following IgG antibody responses to single vaccine brands (5–8). This study is examining how antibody responses vary by vaccine brand, dosage, and days since vaccination, for a period of 1 year minimum.

This is the report on the first 3 months of anti-SARS-2-CoV-spike protein IgG antibodies in a cohort of clinicians. Over at least a 1-year period, we will collect three additional time points in an attempt to understand the clinical relevance of antibody levels over time.

## RESULTS

Among 656 clinicians who consented to participate, 611 (93.1%) completed their baseline survey and serum collection. The mean age of participants was 47.4 years. Approximately two-thirds were female (Table 1). Participants self-identified as primarily white (49.8%), Asian (44%), and non-Hispanic (96.2%). Of the 611 participants, 551 (90.2%) completed the 3-month follow-up. Of the 551 participants, 532 (96.6%) tested negative for SARS-CoV-2 IgG at baseline and therefore were found eligible for seroconversion. Of the 532 participants, 217 (40.8%) received BNT162b2 and 315 (59.2%) received mRNA-1273. The difference in the size of the groups receiving the two vaccines is explained by the fact that this was not a randomized controlled trial but a measurement of the real-world implementation of vaccine rollout in a clinician population. Clinicians went to the closest available hospital or clinic and were given whatever vaccine was available there, which contributed to the slightly uneven numbers.

Seroconversion was demonstrated in 530 (99.6%) of 532 participants. Two participants did not seroconvert following their second dose. In the first nonseroconverting participant, who was receiving a monoclonal antibody (rituximab) against CD20, SARS-CoV-2 antibodies were not detected 28 days following the second dose (BNT162b2) (9). In the second nonseroconverting participant, who was receiving an agent (fingolimod-phosphate) that blocks lymphocytes' ability to emerge from lymph nodes, SARS-CoV-2 antibodies were not detected 21 days following the second dose (mRNA-1273) (10).

Figure 1 depicts the SARS-CoV-2 antibody levels for participants who provided serum samples following vaccination. After adjustments for age and gender, mixed-effect linear (MEL) regression modeling found that the IgG increased significantly after the second dose of vaccine compared to the first dose ($P < 0.001$). Overall, titers peaked at week 6 for both vaccines. Significant differences in IgG antibody levels were found between vaccine brands, higher for mRNA-1273 on days 14 to 20 ($P < 0.05$), 42 to 48 ($P < 0.01$), 70 to 76 ($P < 0.05$), and 77 to 83 ($P < 0.05$) and higher for BNT162b2 on

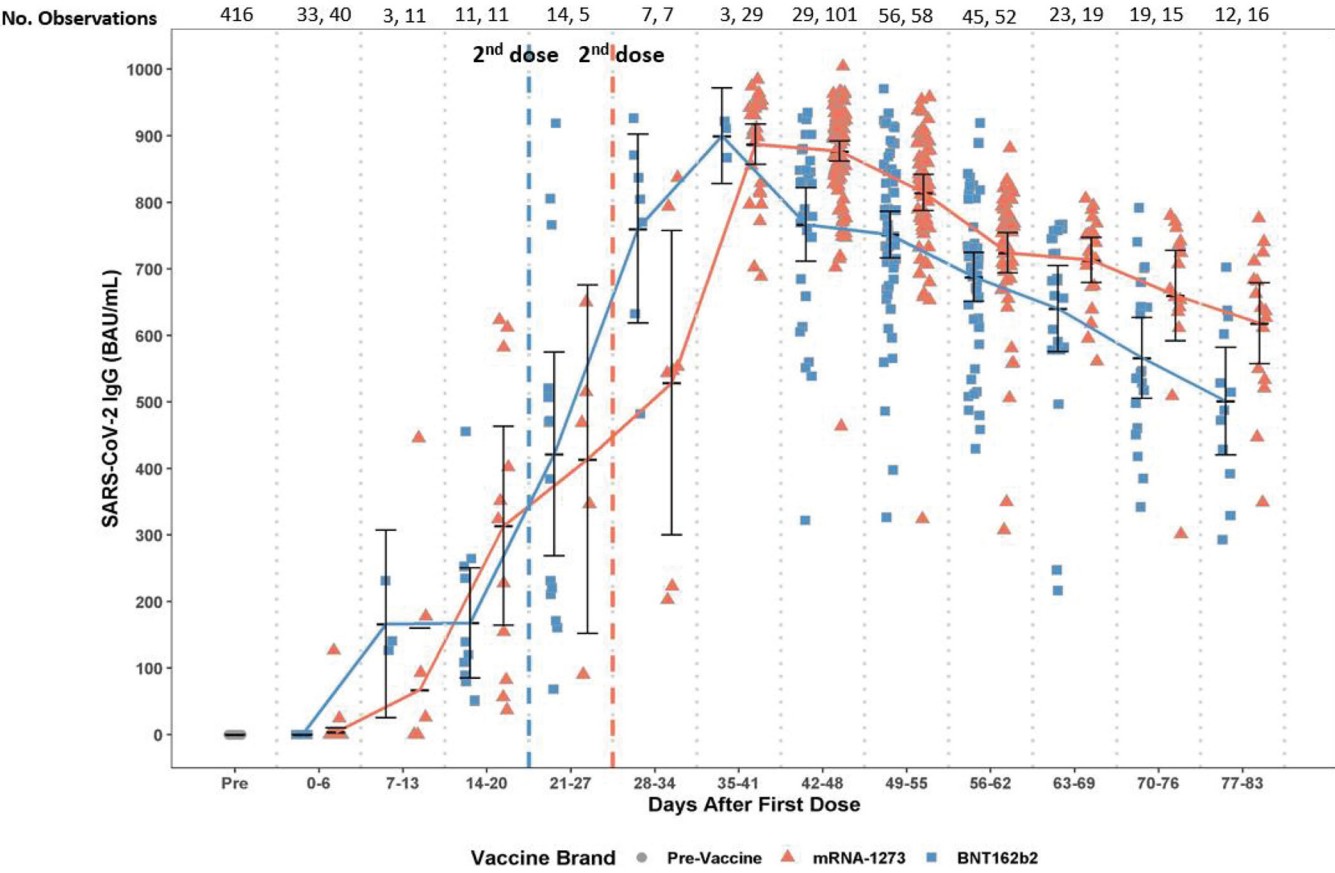

**FIG 1** Levels of SARS CoV-2 IgG against the spike protein's receptor-binding domain by vaccine brand (BNT162b2, mRNA-1273) and over time following the first and second doses.

days 28 to 34 ($P < 0.001$). See Table 2 for the median day and mean IgG titers following the first and second doses of mRNA-1273 and BNT162b2.

During days 0 to 6 postvaccination, 2 of 44 participants who received mRNA-1273 had detectable antibodies. In contrast, during the same period, none of the 33 participants who received BNT162b2 had detectable antibodies.

## DISCUSSION

We detected SARS-CoV-2 IgG seroconversion using an assay aimed at the spike protein receptor binding domain (RBD), in all clinicians following either vaccine, with two exceptions, who were under immunosuppression. Several differences were identified in the IgG responses to BNT162b2 and mRNA-1273. The IgG responses were more robust to mRNA-1273 than to BNT162b2 following the second dose. It is possible that the higher antigenic load delivered by mRNA-1273, containing more than 3-fold the amount of mRNA compared to BNT162b2, explains the significant differences in the observed IgG responses. The fact that the antibody kinetics correlated directly with the number of days since vaccination, booster dose, and quantity of mRNA content suggests that the mRNA vaccine platforms are suitable for the delivery of accurate amounts of antigen, despite the fact that it involves translation steps from RNA to protein.

BNT162b2 is boosted 1 week earlier than mRNA-1273 (resulting in higher levels of antibodies on days 28 to 34), which may explain why the differences in responses were not even wider in favor of mRNA-1273. If protection requires maintaining antibody levels above a certain threshold, higher initial levels of response following vaccination or frequent boosting may succeed in keeping antibody levels above this threshold for longer than after natural infection (11) despite similar rates of antibody decay.

**TABLE 2** Days since first dose and anti-SARS-CoV-2 IgG titers for mRNA-1273 and BNT162b2 following the first and second doses[a]

| No. of days since first dose (range) | mRNA-1273 | | BNT162b2 | |
|---|---|---|---|---|
| | No. of days since first dose (median) | Mean IgG (BAU/mL) | No. of days since first dose (median) | Mean IgG (BAU/mL) |
| 0–6 | 0 | 3.78 | 0 | 0 |
| 7–13 | 9 | 67.50 | 10 | 166.23 |
| 14–20 | 16 | 313.69[b] | 18 | 167.91 |
| 21–27 | 23 | 414.12 | 24.5 | 421.76 |
| 28–34 | 33 | 528.55 | 33 | 760.43[b] |
| 35–41 | 39 | 887.50 | 38 | 899.87 |
| 42–48 | 45 | 876.96[b] | 47 | 766.75 |
| 49–55 | 52 | 814.58 | 51 | 751.41 |
| 56–62 | 58 | 724.21 | 59 | 687.90 |
| 63–69 | 65 | 713.13 | 66 | 640.46 |
| 70–76 | 72 | 659.97[b] | 74 | 566.27 |
| 77–83 | 79.5 | 618.40[b] | 79 | 501.30 |

[a]BNT162b2 delivers 30 $\mu$g of mRNA and is boosted 3 weeks after the first dose. mRNA-1273 delivers 100 $\mu$g of mRNA and is boosted 4 weeks after the first dose. BAU/mL, binding antibody units/mL.
[b]Titers were significantly higher for this vaccine at that time point.

Moreover, even small differences in antibody titers may translate into wider divergence of protection due to amplifiable immune cascades (12).

The limitations of our paper include that we do not yet have the clinical correlates of immunity. We do not know at this time how anti-SARS-CoV-2 IgG antibody titers translate into clinical protection. However, we hope to accrue longitudinally, over a minimum of a 1-year follow-up, clinical correlates of anti-SARS-CoV-2 IgG antibody titers. Additionally, the clinicians who did not seroconvert due to immunosuppression did not have measures of T-cell-mediated immunity that could still be providing protective immune responses (13). Although the difference between the two vaccines we found on days 0 to 6 may reach statistical significance with a larger sample size, the clinical relevance of this finding would still remain unclear at this time. Lastly, this work does not address the presence of neutralizing antibodies, T-cell responses, or immune responses to other non-mRNA vaccines. However, the SARS-CoV-2 IgG titers against the spike protein reported in our cohort have been converted to the World Health Organization (WHO) international standard (BAU/mL) with the hope that our results can be compared to those of other studies.

SARS-CoV-2 IgG titers against the spike protein are available to clinical laboratories but have not been studied as surrogate markers for immune protection. Measure of quantitative SARS-CoV-2 spike protein IgG responses plotted over time following immunization in specific cohorts, while tracking clinical correlates, may help to identify individuals who have titer levels that become nonprotective. This strategy may serve as the basis to have them studied with other correlates of immune protection (e.g., T cells) (13) or be candidates for additional doses. To achieve these goals (11, 13), only serological assays targeting the spike protein and with demonstrated sensitivity and specificity, such as the one used for our study, ought to be utilized. Ongoing studies such as ours can potentially unveil differences in IgG responses between vaccine brands (as observed in this interim report) that may be relevant clinically or for manufacturing purposes (e.g., choice of mRNA amount).

## MATERIALS AND METHODS

A longitudinal study was initiated to estimate the incidence of SARS-CoV-2 infection and COVID-19 by serological testing. Additionally, it aims to assess SARS-CoV-2 antibody responses and sustainability following infection or immunization. Here, we report IgG responses following immunization within the first 3 months of the study. The study protocol was approved by the Sutter Health institutional review board (IRB).

**Serological assay.** The serum samples were not frozen and were analyzed as they were drawn. The serum SARS-CoV-2 IgG was measured using an automated method (Vidas SARS-CoV-2 IgG, bioMérieux, France) with an enzyme-linked fluorescent assay (ELFA) (14) at the Dr. Jack S. Remington Laboratory for Specialty Diagnostics at Sutter Health (hereafter Remington lab, https://www.sutterhealth.org/services/lab-pathology/

toxoplasma-serology-laboratory). Vidas SARS-CoV-2 detects IgG against the receptor binding domain (RBD) of the spike protein. Quantitative results are reported as an index ($\geq$1.00 = positive) (14). Data from the manufacturer and the Remington lab (data not shown) ($n$ = 199) revealed that this assay had a sensitivity of 100% for specimens obtained $\geq$15 days following the onset of symptoms in COVID-19-positive patients. In 989 prepandemic samples from the manufacturer, only one tested positive (99.9% specificity) (14, 15). In an effort to comply with the WHO call for harmonization of SARS-CoV-2 IgG test results (16, 17), we converted the VIDAS SARS-COV-2 IgG index units to binding antibody units (BAU)/mL, where the Vidas SARS-CoV-2 IgG index = 1 (cutoff) = 20.33 BAU/mL (C. A. Gall, personal communication).

**Participants.** A total of 1,769 clinicians were invited to participate and had to sign the informed consent before enrolling via REDCap. The clinicians belong to a multispecialty practice comprised of adult and pediatric primary care physicians, specialists (including hospitalists), and advanced practice clinicians. In addition to completing surveys, the participants provide serum at baseline and every 3 months for a year.

**Statistical analysis.** Mixed-effect linear (MEL) regression modeling was used to examine whether the SARS-CoV-2 IgG index measured over time differed by vaccine brand, dosage, or days since vaccination and to examine the interaction effect between the vaccine brand and number of days since vaccination for the IgG trajectory across time. Modeling adjusted for age and gender and included a subject-specific random intercept term to account for the within-person correlation of measurements over time. The restricted maximum likelihood (REML) approach was used to fit the MEL to produce unbiased estimates of standard errors. MEL regression modeling can handle imbalanced data structure and explicitly model heterogeneity at the individual level and at the occasion level simultaneously by REML (18). Participants who tested positive for SARS-CoV-2 PCR and/or IgG before vaccination ($n$ = 19) were excluded from the model for this report. No participants had "breakthrough infections" (defined as any positive PCR test after the first dose) within 3 months of the study initiation.

In summary, our statistical analysis and conclusions took into account sampling size differences between the two vaccines (mRNA-1273 and BNT162b2), the number of dosages, and the timing of blood draws.

## ACKNOWLEDGMENTS

We thank the clinicians who volunteered and are participating in our study and the Palo Alto Medical Foundation/Bay Area Medical Foundation/Sutter Health for administrative support.

We thank the Palo Alto Medical Foundation donors for financial support.

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
