## [Reviewer comments · Microbiology Spectrum]

Microbiology Spectrum

Differences in IgG antibody responses following BNT162b2 and mRNA-1273 SARS-CoV-2 Vaccines

Jose Montoya, Amy Adams, Valerie Bonetti, Sien Deng, Nan Link, Suzanne Pertsch, Kjerstie Olson, Martina Li, Ellis Dillon, and Dominick Frosch

Corresponding Author(s): Jose Montoya, Palo Alto Medical Foundation

Review Timeline:

Submission Date:	August 4, 2021
Editorial Decision:	September 27, 2021
Revision Received:	October 4, 2021
Accepted:	October 20, 2021

Editor: Daniel Perez

Reviewer(s): The reviewers have opted to remain anonymous.

Transaction Report:

DOI: <https://doi.org/10.1128/Spectrum.01162-21>

September 27, 2021

Prof. Jose Gilberto Montoya
Palo Alto Medical Foundation
Toxoplasma Serology Laboratory, National Reference Center for the Study and Diagnosis of Toxoplasmosis
Palo Alto, California

Re: Spectrum01162-21 (Differences in IgG antibody responses following BNT162b2 and mRNA-1273 Vaccines)

Dear Prof. Jose Gilberto Montoya:

Dear Dr. Montoya,

I apologize for the delay in the review process. It's been extremely difficult to get reviewers for your paper. I have decided to rely on the comments of a single reviewer (attached). Please note that this is very unusual and exceptional. I believe the modifications requested are minor, but I do agree with the comments from the reviewer that you would have a much more impactful story if you were willing to allow more time on the study. However, I do understand the urgency for publishing these initial findings.

Thank you for submitting your manuscript to Microbiology Spectrum. When submitting the revised version of your paper, please provide (1) point-by-point responses to the issues raised by the reviewers as file type "Response to Reviewers," not in your cover letter, and (2) a PDF file that indicates the changes from the original submission (by highlighting or underlining the changes) as file type "Marked Up Manuscript - For Review Only". Please use this link to submit your revised manuscript - we strongly recommend that you submit your paper within the next 60 days or reach out to me. Detailed information on submitting your revised paper are below.

Link Not Available

Sincerely,

Daniel Perez

Journals Department
Reviewer comments:

Reviewer #1 (Comments for the Author):

The manuscript by Montoya et al has assessed IgG antibody titers in a cohort of health care workers vaccinated with 1 of 2 mRNA vaccines. In this initial phase of their study, they present results 3 months post vaccination and conclude that statistically higher immune response is observed for the mRNA-1273 vaccine. The authors have addressed prior reviewer comments and updated the manuscript accordingly.

Major comments:

1) This is the initial phase of a planned > 1 year study. The conclusions from the study would have higher impact if the authors waited to have the full 1 yr data set, specifically in regards to the overall IgG decay in their cohort.

2) Did the authors assess for COVID infection of participants throughout the duration of they study, or just prior to vaccination? Was it possible infection during the study impacted observed antibody titers?

3) Line 48. The authors state that "mRNA-1273 elicited both earlier and higher IgG antibody responses . . ." The authors did demonstrate statistically that higher overall Aby response was observed with 1273; however the data doesn't support that statistically 1273 had earlier response (only 2 subjects were positive and not statistically different). This statement should be modified.

4) Line 62. Please clarify that the vaccine doesn't deliver a higher amount of spike protein. It delivers a higher level of mRNA that is translated into spike protein. Please clarify other instances similar to this throughout the manuscript.

5) Line 97. Please add reference for Remington Lab data or state data no shown.

6)Line 191. Did the authors evaluate T-cell immunity in this cohort? If so, that would be valuable data to include. Minimally, this should be indicated as "data not shown".

Staff Comments:

Preparing Revision Guidelines

Please return the manuscript within 60 days; if you cannot complete the modification within this time period, please contact me. If you do not wish to modify the manuscript and prefer to submit it to another journal, please notify me of your decision immediately so that the manuscript may be formally withdrawn from consideration by Microbiology Spectrum.

Dr. Jack S. Remington Laboratory
for Specialty Diagnostics, a reference
center
for the study and diagnosis of
Toxoplasmosis.
795 El Camino Real, Ames Building
Palo Alto, CA 94301
650-853-4828
Remingtonlab@pamf.org

September 28, 2021

Dear Dr. Perez and Microbiology Spectrum Staff:

Below please find our point-by-point responses to the issues raised by the reviewers
In an attempt to more clearly present our answers to each comment, we have repeated here
each comment and kept the same order in which they appeared in the e-mail sent to us by
Microbiology Spectrum.

In the revised manuscript we have underlined the text that has been modified or added.

Reviewer #1 (Comments for the Author):

The manuscript by Montoya et al has assessed IgG antibody titers in a cohort of health care workers vaccinated with 1 of 2 mRNA vaccines. In this initial phase of their study, they present results 3 months post vaccination and conclude that statistically higher immune response is observed for the mRNA-1273 vaccine. The authors have addressed prior reviewer comments and updated the manuscript accordingly.

Answer: Thank you. As you know our study addressed the immune responses to BNT162b2 and mRNA-1273 Vaccines and as you point, there was statistically higher immune response with the mRNA-1273 vaccine.

Major comments:

1) This is the initial phase of a planned > 1 year study. The conclusions from the study would have higher impact if the authors waited to have the full 1 yr data set, specifically in regards to the overall IgG decay in their cohort.

Answer: We agree with the reviewer that conclusions from our study will eventually have higher impact when our 1 year follow-up results become available. We still believe that there is merit in publishing our early results since they support the results of two other studies that were published despite that they are within the first 3 months following vaccination (Richards NE et. al. JAMA Network Open. 2021;4(9):e2124331. doi:10.1001/jamanetworkopen.2021.24331; Steensels D et. al. JAMA Published online August 30, 2021). However, there is an urgency in publishing our results because our study is the first in utilizing a longitudinal design (the other two studies used a cross-sectional design). Our study is the first to clearly show the differences in immune responses between the two vaccines over time. In addition, our study also revealed the rate of seroconversion among uninfected individuals (“Seroconversion was demonstrated in 530 (99.6%) of 532 participants”, stated in the abstract section, line 39-40).

2) Did the authors assess for COVID infection of participants throughout the duration of they study, or just prior to vaccination? Was it possible infection during the study impacted observed antibody titers?

Answer: Yes, COVID infection was assessed by participants self-report. Participants were asked at the time of enrollment in the study about any symptoms and PCR test results in the prior three months. Infection was defined as a positive PCR test result anytime or a positive antibody test prior to vaccination. If the positive PCR or IgG antibody test was before the 1st vaccine dose, we considered it as ‘pre-vaccine infection’. If the positive PCR test was after 1st dose, we considered it as ‘breakthrough infection’. Based on the data presented in the manuscript, 19 subjects were identified with ‘pre-vaccine infection’. These 19 participants were excluded from our analysis. Please see our statement in the original manuscript: “Participants who tested positive for SARS-CoV-2 PCR and/or IgG before vaccination (n = 19) were excluded from the model for this report” (lines 118-119).

No participants had “breakthrough infections” (defined as any positive PCR test after the 1st dose) within three months of study initiation. We have added this statement to the

Statistical analysis section (lines 119-121 in the marked-up manuscript). Thank you for raising this point.

3) Line 48. The authors state that "mRNA-1273 elicited both earlier and higher IgG antibody responses . . ." The authors did demonstrate statistically that higher overall Aby response was observed with 1273; however the data doesn't support that statistically 1273 had earlier response (only 2 subjects were positive and not statistically different). This statement should be modified.

Answer: We see the reviewer's point. We have now deleted reference to earlier antibody response from the abstract (Line 48 in the original manuscript, see delete in line 46 of revised manuscript) and in the discussion section (Lines 169-172 in the original manuscript, delete in lines 161-164 in revised manuscript).

4) Line 62. Please clarify that the vaccine doesn't deliver a higher amount of spike protein. It delivers a higher level of mRNA that is translated into spike protein. Please clarify other instances similar to this throughout the manuscript.

Answer: The reviewer is correct. We apologize for the confusion created by our wording, Thank you. We have now corrected this throughout the manuscript (Lines 33-34, 59-60, 66-69, 165-170, 201, and Table 2 footnote).

5) Line 97. Please add reference for Remington Lab data or state data no shown.

Answer: We have now added a statement indicating "data not shown" (Line 94). Thank you.

6) Line 191. Did the authors evaluate T-cell immunity in this cohort? If so, that would be valuable data to include. Minimally, this should be indicated as "data not shown".

Answer: Unfortunately, we did not evaluate T-cell mediated immunity. We have added this to our list of limitations (Line 186-188).

Sincerely,

José G. Montoya M.D., FACP, FIDSA
Dr. Jack S. Remington Laboratory for Specialty Diagnostics
National Reference Laboratory for the Study and Diagnosis of Toxoplasmosis and special
pathogens.
Palo Alto Medical Foundation
795 El Camino Real
Ames Building
Palo Alto, CA, USA
E-mail: montoyj@sutterhealth.org

October 20, 2021

Prof. Jose Gilberto Montoya
Palo Alto Medical Foundation
Toxoplasma Serology Laboratory, National Reference Center for the Study and Diagnosis of Toxoplasmosis
Palo Alto, California

Re: Spectrum01162-21R1 (Differences in IgG antibody responses following BNT162b2 and mRNA-1273 SARS-CoV-2 Vaccines)

Dear Prof. Jose Gilberto Montoya:

Your manuscript has been accepted, and I am forwarding it to the ASM Journals Department for publication. You will be notified when your proofs are ready to be viewed.

Sincerely,

Daniel Perez
Editor, Microbiology Spectrum
